# Association Between Previous Statin Use and Alzheimer’s Disease: A Nested Case-Control Study Using a National Health Screening Cohort

**DOI:** 10.3390/brainsci11030396

**Published:** 2021-03-20

**Authors:** Ji Hee Kim, Heui Seung Lee, Jee Hye Wee, Yoo Hwan Kim, Chan Yang Min, Dae Myoung Yoo, Hyo Geun Choi

**Affiliations:** 1Department of Neurosurgery, College of Medicine, Hallym University, Anyang 14068, Korea; kimjihee.ns@gmail.com (J.H.K.); seunglee@hallym.or.kr (H.S.L.); 2Department of Otorhinolaryngology, College of Medicine, Hallym University, Anyang 14068, Korea; weejh07@hanmail.net; 3Department of Neurology, College of Medicine, Hallym University, Anyang 14068, Korea; drneuroneo@gmail.com; 4Hallym Data Science Laboratory, College of Medicine, Hallym University, Anyang 14068, Korea; joicemin@naver.com (C.Y.M.); ydm1285@naver.com (D.M.Y.)

**Keywords:** Alzheimer’s disease, cholesterol, dementia, neurodegenerative dementia, risk factor, statin

## Abstract

A number of studies report the incidence of Alzheimer’s disease (AD) in patients taking statins, but the results are inconsistent. (1) Background: The present study investigated the cross-sectional association between previous statin use and the risk of AD development in Korean residents. (2) Methods: We used the Korean National Health Insurance Service-National Sample Cohort; 17,172 AD patients were matched by age, gender, income, and region of residence with 68,688 control participants at a ratio of 1:4. We used a multiple conditional logistic regression model to analyse the association between the number of days of statin use and AD occurrence. Further analyses were performed to identify whether this association is maintained for different ages, genders, socioeconomic status groups, and covariates. (3) Results: The odds ratio, which was adjusted for potential confounders, for the days of statin use per year in the AD group compared to the control group was 0.95 (95% confidence interval = 0.92–0.98; *p* = 0.003). The number of days of statin use in the AD group was significantly smaller in the subgroups of non-smokers and individuals with normal weight, alcohol consumption less than once a week, total cholesterol level below 200 mg/dL, systolic blood pressure below 140, diastolic blood pressure below 90, and fasting blood glucose below 100 mg/dL. (4) Conclusions: Our results suggest that statin use prevents the occurrence of AD. The effects of statin use in preventing AD may be greater in individuals at relatively low risk.

## 1. Introduction

Alzheimer’s disease (AD) is the most common form of dementia and the most common cause of neurodegenerative dementia in elderly individuals, accounting for approximately 60% or more of all individuals with dementia [1]. Over 50 million people worldwide are living with dementia, and the number of patients affected is expected to rise to 152 million by 2050 [2]. This rapidly increasing disease burden is challenging healthcare and socioeconomic systems worldwide. Despite substantial research and development investment in AD, none of the currently available medications prevents AD or modifies its pathology [3]. Therapeutic approaches for disease-modifying effects target the amyloid cascade to prevent the accumulation of toxic amyloid aggregates [4]. However, recent clinical trials in AD, many based on the amyloid hypothesis, were disappointing, and this failure has increased the interest in other potential therapeutic avenues [5].

Statins are the most widely prescribed Food and Drug Administration (FDA)-approved cholesterol-lowering medications [6]. Statins are 3-hydroxy-3-methylglutaryl coenzyme A reductase inhibitors that reduce cholesterol synthesis and upregulate low-density lipoprotein (LDL) receptors in hepatocyte membranes, which enables greater clearance of LDL from the bloodstream and lowers the level of lipoproteins [2]. Statins demonstrate various additional effects, including antioxidant, anti-inflammatory, antithrombotic, anti-excitotoxicity, and angiogenesis-promoting effects, improvements in vascular endothelial function, and the modulation of amyloid and tau metabolism [7,8,9,10]. Therefore, several studies have examined the possible benefits of statins in preventing or reducing the risk of AD and dementia [9,11,12]. Among all causes of dementia, vascular dementia is more likely to benefit from statins use because it shares the same risk factors as ischemic stroke, and elevated levels of serum total cholesterol and LDL and decreased levels of high-density lipoprotein (HDL) are associated with the risk of vascular dementia [13]. However, the role of statins in the improvement of cognitive function and the prevention of AD is controversial.

Accordingly, the present study examined the association between previous statin use and the incidence of AD while adjusting for putative risk factors for AD occurrence. We also assessed whether these associations are consistent for different groups in a subgroup analysis stratified by age, gender, socioeconomic factors such as income and region of residence, and baseline medical comorbidities including dyslipidaemia, total cholesterol, obesity, smoking status, alcohol consumption, blood pressure, and fasting blood glucose.

## 2. Materials and Methods

### 2.1. Study Population

The Institutional Review Board (IRB) of the ethics committee of Hallym University approved this study (IRB No. 2019-10-023). The need to obtain informed consent was waived because of the use of deidentified claims data. All analyses adhered to the guidelines and regulations of the ethics committee of Hallym University. A detailed description of the Korean National Health Insurance Service (NHIS)-Health Screening Cohort data has been delineated elsewhere [14].

### 2.2. Patient Selection

AD patients were selected from a total of 514,866 participants with 615,488,428 medical claim codes (*n* = 20,087). The control group comprised all participants who were not dementia patients (*n* = 494,779). To include AD patients who were diagnosed for the first time, we excluded AD patients who were diagnosed from 2002 to 2003 (washout period, *n* = 407). Participants were excluded if they were <60 years old (*n* = 661 for AD patients, *n* = 179,499 for control participants). Control participants with ICD-10 codes G30 or F00 from 2002 through 2015 were excluded (*n* = 5209). Participants for whom total cholesterol records (*n* = 15 for dementia patients), blood pressure records (*n* = 2 for AD patients), fasting blood glucose records (*n* = 1 for AD patients), or body mass index (BMI, kg/m^2^) records (*n* = 2 for AD patients) were not available were excluded. 

AD patients were matched with control participants at a ratio of 1:4 for age, gender, income, and region of residence. To curtail selection bias, the control participants were selected in random number order. The index date of each AD patient was set as the time of treatment for dementia. The index date of each control participant was fixed as the index date of their matched AD patient. Therefore, each matched AD patient and control participant had the same index date. During the matching process, 1827 AD patients and 241,383 control participants were excluded. Ultimately, 17,172 AD patients were matched with 68,688 control participants (ratio of 1:4) (Figure 1).

### 2.3. The Number of Days of Statin Use (Exposure)

The sum of the total days of statin prescription was continuously counted for 2 years (730 days) before the index date. Atorvastatin, fluvastatin, lovastatin, pitavastatin, pravastatin, rosuvastatin, and simvastatin were included.

### 2.4. Alzheimer’s Disease (Outcome)

Participants had AD if they were diagnosed with ICD-10 code G30 or F00 (dementia in Alzheimer’s disease). To ensure the accuracy of the diagnosis, we selected only participants who were treated ≥2 times [15,16]. We provide a description of the reliability of the diagnosis of dementia in Appendix A.

### 2.5. Covariates

Age groups were divided into 5-year intervals: 60–64, 65–69, 70–74…, and 85+ years old (total of six age groups). Income groups were classified into five classes (class 1 [lowest income] to 5 [highest income]). The region of residence was sorted into urban and rural areas, as described in our previous research [17]. Tobacco smoking, alcohol consumption, and obesity (determined using BMI) were categorized in the same manner as our previous study [18]. Systolic blood pressure (SBP), diastolic blood pressure (DBP), fasting blood glucose, and total cholesterol were measured. A participant had dyslipidaemia if an ICD-10 code of E78 was present ≥2 times before the index date. The Charlson Comorbidity Index (CCI) was used extensively to assess disease burden using 17 comorbidities as the continuous variable (0 [no comorbidities] to 29 [multiple comorbidities]) [19]. CCI scores were calculated for all comorbidities except dementia.

### 2.6. Statistical Analyses

The general characteristics of the AD and control groups were compared using the chi-square test for categorical variables and the independent *t* test for continuous variables.

To analyse the odds ratios (ORs) with 95% confidence intervals (CIs) for the days of statin use per year in AD patients compared with control participants, conditional logistic regression was used. In this analysis, we examined the crude model, model 1 (adjusted for dyslipidaemia history, total cholesterol, SBP, DBP, and fasting blood glucose) and model 2 (adjusted for model 1 covariates plus obesity, smoking, alcohol consumption, and CCI scores). The analysis was stratified by age, gender, income (lower income [1,2,3] and high income [4,5]), and region of residence.

For the subgroup analyses, we divided participants by age and gender (<75 years old and ≥75 years old; men and women, respectively) and analysed the crude model, model 1, and model 2. Another subgroup analysis based on obesity status (underweight, normal weight, overweight, and obese [obese I and II]) was performed using unconditional logistic regression. In these analyses, model 1 (adjusted for age, gender, income, and region of residence), model 2 (adjusted for model 1 plus dyslipidaemia history, total cholesterol, SBP, DBP, and fasting blood glucose), and model 3 (adjusted for model 2 plus smoking, alcohol consumption, and CCI scores) were calculated. Other subgroup analyses were similarly performed on the basis of smoking status (non-smoker and past/current smoker), alcohol consumption (<1 time a week and ≥1 time a week), total cholesterol (<200, ≥200 to <240, and ≥240), blood pressure (SBP <140 and DBP <90, and SBP ≥140 and DBP ≥90), and fasting blood glucose (<100 and ≥100). Models 1–3 were also calculated in the same manner in these analyses. We added the *p*-values of interaction between exposure and covariates for outcome.

All statistical analyses were performed using SAS version 9.4 (SAS Institute Inc., Cary, NC, USA). Two-tailed analyses were performed, and differences with probability values <0.05 were considered statistically significant.

## 3. Results

The general characteristics (age, gender, income, and region of residence) of participants did not differ between the two groups because of the matching procedures (*p* = 1.000). Mean total cholesterol level, mean DBP, and the mean number of days of statin prescription did not significantly differ between the AD and control groups (*p* = 0.284, *p* = 1.000, and *p* = 0.997, respectively). However, the mean SBP in the AD group (131.1 mmHg) was lower than in the control group (131.4 mmHg; *p* < 0.022). The mean fasting glucose level (107.5 mg/dL) and the rate of dyslipidaemia (37.8%) in the AD group were higher than in the control group (103.1 mg/dL; 36.1%; *p* < 0.001). The rate of obesity, smoking status, alcohol consumption, CCI scores, and the period of statin prescription significantly differed between the AD and control groups (*p* < 0.001). The demographic and clinical characteristics are summarized in Table 1.

The crude OR, which was adjusted only for age, gender, income, and region of residence, and the adjusted OR in model 1, which was adjusted for dyslipidaemia history, total cholesterol, SBP, DBP, and fasting blood glucose, for the number of days of statin use per year in the AD group, were 1.00 (95% CI = 0.97–1.03; *p* = 0.997) and 0.96 (95% CI = 0.93–0.99; *p* = 0.003), respectively. The adjusted OR in model 2 was significant (0.95; 95% CI = 0.92–0.98; *p* = 0.003), which was adjusted for additional confounders, including obesity, smoking, alcohol consumption, and CCI scores. The crude and adjusted ORs of the number of days of statin use per year in the AD group compared to the control group and subgroup analyses according to age and gender are shown in Table 2.

In the subgroup analyses stratified by age and gender, the adjusted OR in model 2 for the number of days of statin use per year was significantly lower in the AD group among women older than 75 years (adjusted OR in model 2 = 0.92; 95% CI = 0.88–0.97; *p* = 0.002). The adjusted OR in model 2 for the number of days of statin use per year was lower in the AD group in individuals with high income in subgroup analyses stratified for income (adjusted OR in model 2 = 0.95; 95% CI = 0.91–0.99; *p* = 0.013), and it was lower for all participants who lived in urban and rural areas when stratified for the region of residence (adjusted OR in model 2 = 0.95 and 0.96; 95% CI = 0.90–0.99 and 0.92–1.00; *p* = 0.013 and *p* = 0.035, respectively).

For subgroup analyses stratified for obesity, smoking status, alcohol consumption, total cholesterol, blood pressure, and fasting blood glucose, the adjusted OR in model 3 was significantly lower in the AD group than in the control group in the subgroups of non-smokers (adjusted OR in model 3 = 0.96; 95% CI = 0.90–1.00; *p* = 0.011), individuals with normal weight (adjusted OR in model 3 = 0.95; 95% CI = 0.90–1.00; *p* = 0.044), alcohol consumption less than once a week (adjusted OR in model 3 = 0.96; 95% CI = 0.92–0.99; *p* = 0.013), total cholesterol level below 200 mg/dL (adjusted OR in model 3 = 0.94; 95% CI = 0.90–0.98; *p* = 0.003), SBP below 140 and DBP below 90 mmHg (adjusted OR in model 3 = 0.96; 95% CI = 0.92–0.99; *p* = 0.019), and fasting blood glucose below 100 mg/dL (adjusted OR in model 3 = 0.94; 95% CI = 0.90–0.98; *p* = 0.008; Figure 2; Appendix A).

## 4. Discussion

This nested case-control study investigated the association between the number of days of statin use and the occurrence of AD using age-, gender-, income-, and region-of-residence-matched cohorts. The main findings of this study demonstrated that the number of days of statin use per year in the AD group was significantly lower than in the control group, with an adjusted OR of 0.95 (95% CI = 0.92–0.98; *p* = 0.003). This difference was maintained in the subgroup of women older than 75 years. We did not find any differences in the subgroup of men regardless of age and the subgroup of women younger than 75 years when stratified by age and gender. Further stratified analyses of putative risk factors, we found that the association between the number of days of statin use and the occurrence of AD was more prominent in non-smokers, individuals with normal weight, alcohol consumption less than once a week, total cholesterol level below 200 mg/dL, SBP below 140 and DBP below 90, and fasting blood glucose below 100 mg/dL. These results suggest that the role of statins in the prevention of AD is more pronounced in relatively low-risk groups than in high-risk groups of AD.

Over the past few years, a large number of observational studies reported a significant association between statin use and a reduced risk of dementia [20,21,22,23,24], but two randomized controlled trials (RCTs) did not support the hypothesis that statins prevented AD. The Medical Research Council (MRC) and the British Heart Foundation (BHF) Heart Protection Study of 20,536 participants with increased risk for vascular events evaluated the effect of simvastatin on dementia risk and showed no benefit for dementia prevention [11]. The Prospective Study of Pravastatin in the Elderly at Risk (PROSPER) trial of 5804 participants aged 70–82 years with a history of, or risk factors for, vascular disease found that pravastatin had no significant effect on cognitive function or disability compared with placebo [12]. A Cochrane Database systematic review of double-blind, randomized, placebo-controlled trials in which statins were administered for at least 12 months to individuals at risk of AD and dementia analysed two trials with 26,340 participants aged 40 to 82 years [9]. This research reported no reduction in the occurrence of AD or dementia in individuals treated with statins compared with those given placebo and concluded that statins given in late life to those at risk of vascular disease played no role in preventing cognitive decline or dementia. In contrast, two meta-analyses yielded conflicting results. One systematic review and meta-analysis of 16 studies using qualitative synthesis and 11 using quantitative synthesis examined short- and long-term cognitive effects of statins on patients without a history of cognitive dysfunction and revealed a 29% reduction in the incidence of dementia in statin-treated patients (hazard ratio = 0.71; 95% CI = 0.61–0.82) [25]. Another meta-analysis of 25 RCTs reported no effect of statins on cognitive function in a total of 46,836 cognitively healthy and impaired participants [26]. Although many studies on the effectiveness of statins in the treatment or prevention of AD showed inconsistent results, our study supports the hypothesis that statin use contributes to preventing the occurrence of AD. The effect of this prevention was greater in individuals with a lower risk of developing dementia than in individuals with a high risk of developing dementia.

The mechanism underlying the association between statin treatment and a reduced risk of AD may be explained by the multiple pleiotropic effects of statins. First, statins decrease the inflammatory process, which has beneficial effects in reducing the risk of dementia [27]. Animal models and clinical studies strongly suggest that inflammation contributes to AD pathogenesis [28]. Statins produce an overall anti-inflammatory effect by preventing the release of proinflammatory chemokines, matrix metalloproteinases (MMPs), such as MMP-2, MMP-7, and MMP-9, cytokines, including tumour necrosis factor (TNF)-α, interleukin (IL)-1β and IL-6, and adhesion molecules from inflammatory cells [29,30]. Second, statins may reduce the production of β-amyloid protein by inhibiting cholesterol biosynthesis to decrease amyloid production [31,32,33]. β-Amyloid protein is a neurotoxin and a proteolytic fragment of amyloid precursor protein (APP), which plays a major role in the pathogenesis of AD. Third, statins increase the production of vasodilators, such as prostaglandin I2, and the level of endothelial nitric oxide synthase (eNOS), and decrease the production of vasoconstrictors, such as angiotensin II, which further improves cerebral perfusion [34,35]. Anticoagulation, angiogenesis-promoting effects, and improvement in vascular endothelial function may enhance cerebral perfusion [8]. Potential neuroprotective benefits from statins occur from the preservation of the cerebral blood flow, which decreases the risk of AD because AD is closely associated with cerebral hypoperfusion. Fourth, statins have antioxidant properties due to the inhibition of nicotinamide adenine dinucleotide phosphate (NADPH) oxidase, which may contribute to their antioxidant effect [36]. These combined effects may exert the beneficial effect of statins in preventing AD. Many studies demonstrated that statins did not improve cognition for patients who were already diagnosed with dementia. Our study also showed that the preventive effect of statins on AD was apparent in individuals without baseline medical comorbidities.

The strength of the present study is the use of NHIS data, which represents all Korean citizens without exceptions. This cohort is a large, highly representative, nationwide population sample, and there were no missing participants. Due to the substantial number of participants, we randomly selected a control group using 1:4 matching to the AD group by age, gender, income, and region or residence. We also adjusted our analyses for factors affecting the use of statins, such as dyslipidaemia history and total cholesterol, and other potential confounders such as obesity, smoking status, alcohol consumption, blood pressure, fasting blood glucose, and CCI scores.

However, several potential limitations should be considered. First, the diagnosis of AD was based on ICD-10 coding, and there are concerns about the accuracy of medical coding in claims data. To compensate for this limitation, we demonstrated that the coding of AD from NHIS data had good validity in our previous study (see Appendix A). The diagnosis of AD remained based on the clinical history and presentation of the patients as examined by the physician, regardless of the advent of various biomarkers and advanced medical imaging techniques, including molecular imaging. Therefore, the diagnosis of AD mostly included a thorough history and a subsequent cognitive function test, such as the Montreal Cognitive Assessment (MoCA) or Mini Mental State Examination (MMSE), but we could not identify whether clinicians performed such cognitive tests in the Korean NHIS database. Hence, there is a possibility that AD was underestimated or overestimated. 

Second, unmeasured confounders, such as other putative risk factors for AD, could not be analysed in this study. Although approximately 70% of the risk of developing AD may be attributed to genetics, we did not consider genetic risk factors. Well-established acquired factors, such as diabetes, hypertension, obesity, smoking, and dyslipidaemia, were included as confounders. Nevertheless, information on other acquired factors, including marital status, stress, depression, inadequate sleep, physical activity, diet, vitamin D, and hormonal replacement therapy, was not available in this study [37,38]. AD commonly exists with other comorbidities, such as Lewy body dementia and vascular dementia, as a mixed dementia. The Rochester Epidemiology Project with 419 elderly demented patients found that the post-mortem diagnosis of AD was established in 51%, that of pure vascular dementia was found in 13%, and that of mixed vascular-Alzheimer dementia was established in the 12% of patients with “other” diagnosis in the remaining patients [39]. It is evident from autopsy studies that many patients with mixed dementia have vascular and degenerative causes [40,41]. The three main causes of vascular dementia are large-vessel strokes, small-vessel disease, and microhaemorrhages, and the treatment of vascular dementia involves primary and secondary prevention of stroke and cardiovascular disease, which decreases the burden of vascular dementia. Therefore, statins may be effective in preventing primarily vascular dementia by modifying vascular risk factors in mixed dementia. 

Finally, several previous studies suggested that the significant association of statins use with AD was robust or limited to lipophilic statins [21,22]. Statins are divided into two different subclasses according to their solubility (lipophilic or hydrophilic) and according to their origin and their activity (synthetic or naturally occurring products). Thus, the association between the risk of incident dementia may depend on the different levels of lipid control by specific statins and their different properties of blood-brain penetration. However, we did not consider which statins were used as treatment in this study. Further analyses are necessary to determine which statin groups or specific statins are effective in preventing AD.

## 5. Conclusions

Overall, our results indicate the clinical importance of statin use in preventing dementia. The use of statins resulted in greater benefit for women older than 75 years and individuals with a low risk of AD development. Further exploration is warranted to clarify the relationship between statin use and the prevention of AD and thus confirm this hypothesis.

## Figures and Tables

**Figure 1 brainsci-11-00396-f001:**
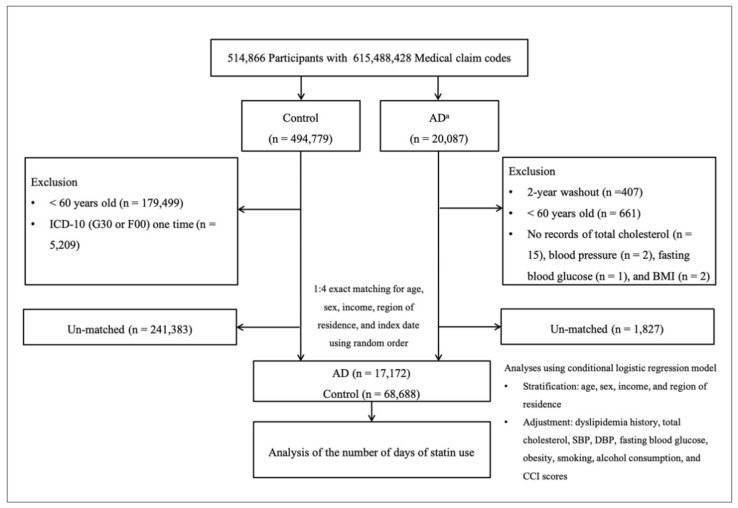
A schematic illustration of the participant selection process used in this study. Of the total 514,866 participants, 17172 AD patients were matched with 68688 control participants for age, gender, income, and region of residence. “a” indicates that AD patients were selected as the participants assigned G30 or F00 based on ICD-10 codes ≥2 times. AD–Alzheimer’s disease; BMI–body mass index; CCI–Charlson Comorbidity Index; DBP–diastolic blood pressure; SBP–systolic blood pressure.

**Figure 2 brainsci-11-00396-f002:**
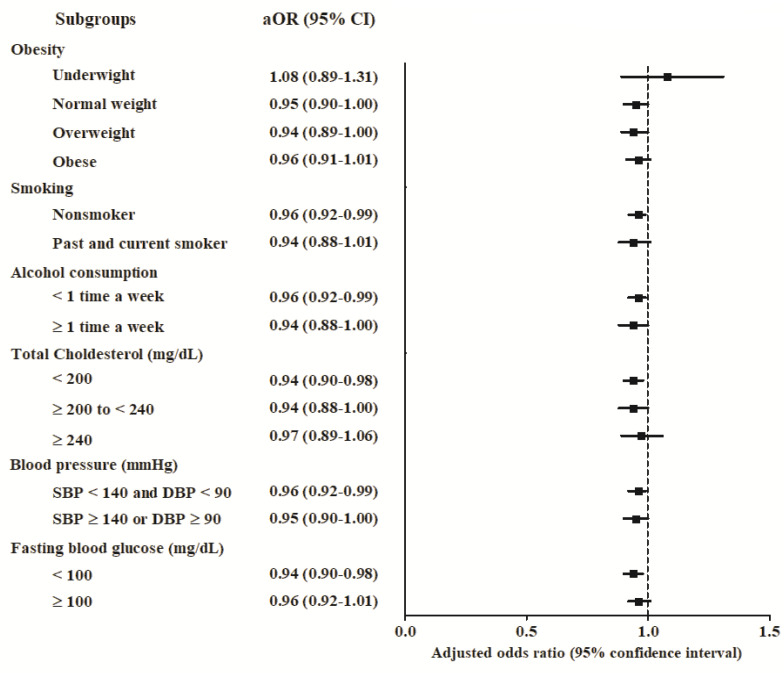
Adjusted odds ratios and 95% confidence intervals for the days of statin use per year in AD patients compared with control participants according to various comorbidities. AD—Alzheimer’s disease; CI—confidence interval; DBP—diastolic blood pressure; OR—odds ratio; SBP–systolic blood pressure.

**Table 1 brainsci-11-00396-t001:** General characteristics of participants.

Characteristics		Total Participants		
		AD	Control	*p* Value
Age (years, *n*, %)				1.000
	60–64	951 (5.5)	3804 (5.5)	
	65–69	2212 (12.9)	8848 (12.9)	
	70–74	4237 (24.7)	16,948 (24.7)	
	75–79	5311 (30.9)	21,244 (30.9)	
	80–84	3826 (22.3)	15,304 (22.3)	
	85+	635 (3.7)	2540 (3.7)	
Gender (*n*, %)				1.000
	Male	6742 (39.3)	26,968 (39.3)	
	Female	10,430 (60.7)	41,720 (60.7)	
Income (*n*, %)				1.000
	1 (lowest)	3519 (20.5)	14,076 (20.5)	
	2	1937 (11.3)	7748 (11.3)	
	3	2324 (13.5)	9296 (13.5)	
	4	3067 (17.9)	12,268 (17.9)	
	5 (highest)	6325 (36.8)	25,300 (36.8)	
Region of residence (*n*, %)				1.000
	Urban	5986 (34.9)	23,944 (34.9)	
	Rural	11,186 (65.1)	44,744 (65.1)	
Total cholesterol (mg/dL, mean, SD)		197.7 (41.5)	197.3 (39.5)	0.284
SBP (mmHg, mean, SD)		131.1 (18.0)	131.4 (17.2)	0.022 ^b^
DBP (mmHg, mean, SD)		78.7 (11.1)	78.7 (10.7)	1.000
Fasting blood glucose (mg/dL, mean, SD)		107.5 (38.6)	103.1 (29.7)	<0.001 ^b^
Obesity (*n*, %) ^c^				<0.001 ^a^
	Underweight	942 (5.5)	2935 (4.3)	
	Normal	6898 (40.2)	24,979 (36.4)	
	Overweight	4053 (23.6)	17,587 (25.6)	
	Obese I	4785 (27.9)	20,969 (30.5)	
	Obese II	494 (2.9)	2218 (3.2)	
Smoking status (*n*, %)				<0.001 ^a^
	Non-smoker	13,587 (79.1)	54,537 (79.4)	
	Past smoker	1715 (10.0)	7570 (11.0)	
	Current smoker	1870 (10.9)	6581 (9.6)	
Alcohol consumption (*n*, %)				<0.001 ^a^
	<1 time a week	13,339 (77.7)	51,085 (74.4)	
	≥1 time a week	3833 (22.3)	17,603 (25.6)	
CCI score (score, *n*, %) ^d^				<0.001 ^a^
	0	6256 (36.4)	38,044 (55.4)	
	1	4068 (23.7)	13,134 (19.1)	
	2	2605 (15.2)	7682 (11.2)	
	3	1945 (11.3)	4459 (6.5)	
	≥4	2298 (13.4)	5369 (7.8)	
Dyslipidaemia (*n*, %)		6485 (37.8)	24,781 (36.1)	<0.001 ^a^
Number of days of statin use per year (days, mean, SD)		114.4 (228.8)	114.4 (234.2)	0.997
Periods of statin prescription (*n*, %)				<0.001 ^a^
	0 days	118,86 (69.2)	49,995 (72.8)	
	≥1 day & <6 months	1838 (18.7)	5104 (7.4)	
	≥6 months & <1 year	877 (5.1)	3175 (4.6)	
	≥1 year & <1.5 years	656 (3.8)	2234 (3.3)	
	≥1.5 years	1915 (11.2)	8180 (11.9)	

Note: AD–Alzheimer’s disease; CCI–Charlson comorbidity index; DBP–diastolic blood pressure; SBP–systolic blood pressure; SD–standard deviation. ^a^ Chi-squared test. Significance at *p* < 0.05; ^b^ Independent *t* test. Significance at *p* < 0.05; ^c^ Obesity (BMI, body mass index, kg/m^2^) was categorized as <18.5 (underweight), ≥18.5 to <23 (normal), ≥23 to <25 (overweight), ≥25 to <30 (obese I), and ≥30 (obese II); ^d^ CCI scores were calculated for all comorbidities except dementia.

**Table 2 brainsci-11-00396-t002:** Odds ratios (95% confidence interval) for the days of statin use per year in Alzheimer’s disease patients compared to control participants, with subgroup analyses according to age, gender, income, and region of residence.

Characteristics	Odds Ratios						*p* Value for Interaction
	Crude ^b^	*p* Value	Model 1 ^bc^	*p* Value	Model 2 ^bd^	*p* Value	
Total participants (*n* = 85,860)	1.00 (0.97–1.03)	0.997	0.96 (0.93–0.99)	0.003 ^a^	0.95 (0.92–0.98)	0.003 ^a^	
Age <75 years old, men (*n* = 14,840)	1.08 (1.01–1.16)	0.019 ^a^	1.00 (0.92–1.08)	0.918	1.00 (0.92–1.08)	0.912	0.009 ^a^
Age <75 years old, women (*n* = 22,160)	1.08 (1.02–1.13)	0.004 ^a^	0.98 (0.92–1.04)	0.420	0.98 (0.92–1.04)	0.487	
Age ≥75 years old, men (*n* = 18,870)	0.98 (0.9–1.04)	0.581	0.96 (0.90–1.03)	0.210	0.95 (0.89–1.02)	0.150	
Age ≥75 years old, women (*n* = 29,990)	0.93 (0.89–0.97)	0.001 ^a^	0.92 (0.88–0.97)	0.002 ^a^	0.92 (0.88–0.97)	0.002 ^a^	
Low income (*n* = 38,900)	1.01 (0.97–1.05)	0.586	0.96 (0.92–1.01)	0.090	0.96 (0.91–1.00)	0.074	0.448
High income (*n* = 46,960)	0.99 (0.96–1.03)	0.650	0.95 (0.91–0.99)	0.014 ^a^	0.95 (0.91–0.99)	0.013 ^a^	
Urban (*n* = 29,930)	0.99 (0.95–1.03)	0.653	0.95 (0.90–0.99)	0.028 ^a^	0.95 (0.90–0.99)	0.029 ^a^	0.795
Rural (*n* = 55,930)	1.01 (0.97–1.04)	0.717	0.96 (0.92–1.00)	0.044 ^a^	0.96 (0.92–1.00)	0.035 ^a^	

Note: CCI—Charlson comorbidity index; DBP—diastolic blood pressure; SBP—systolic blood pressure. ^a^ Conditional logistic regression, significance at *p* < 0.05; ^b^ Models were stratified by age, gender, income, and region of residence; ^c^ Model 1 was adjusted for dyslipidaemia history, total cholesterol, SBP, DBP, and fasting blood glucose; ^d^ Model 2 was adjusted for model 1 covariates plus obesity, smoking, alcohol consumption, and CCI scores.

## Data Availability

The current article used a national sample cohort and does not involve data that can be made available.

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
