# Peer review of "Association between Previous Statin Use and Alzheimer’s Disease: A Nested Case-Control Study Using a National Health Screening Cohort"

_brainsci, 2021, doi:10.3390/brainsci11030396_

Round 1

Reviewer 1 Report

In this manuscript, the authors used nested case-control study to study the association between statins use and the risk of AD development, and found that statins using might be able to prevent the occurrence of AD. This study is very interesting. However, there are still concerns needed to be addressed before the manuscript can be published. It would be necessary to find a native English speaker to check the writing and grammars.

1, The conclusion of this manuscript is ‘The use of statins resulted in greater benefit for women older than 75 years and individuals with a low risk of AD development.’ However, there are few things need to be considered before the conclusion can be made. 1) The statins exposure was counted for 2 years. There would be lots of patients have been taken stains for more than 2 years. May be women > 75 years took statins for a longer time, and then the protective effects of statins showing up? 2) The statins dose is another important factor, which the authors need to think.

2, There are quite a few citations missing. Like introduction section: 1) ‘none of the currently available medications prevent AD or modifies its pathology’. 2) ‘To date, therapeutic approaches for disease-modifying effects have focused on targeting the amyloid cascade, attempting to prevent the accumulation of toxic amyloid aggregates.’ 3) ‘Consequently, several studies have examined the possible benefits of statins in preventing or reducing the risk of AD and dementia.’

3, The World Alzheimer Report 2020 has the most recently AD patient’s number. It would be a better source to show the rapid increasing of AD recently.  

4, The ‘study population’, ‘Alzheimer’s disease’ and ’patient selection’ sections can be reorganized to make it easier to read.

5, Alcohol consumption should be considered as the amount of alcohol intake per week, instead of times/week. For smoking, 1 cigarette/ week and 100 cigarettes/ week are quite different. A definition is needed here. 

Reviewer 2 Report

The study is interesting and the results would be worth publishing. However, some sloppy errors need to be corrected and the statistical analysis, including the associated discussion, should be significantly expanded, in my opinion.

My suggestions:
For me, the introduction is sufficient (but I must say that I am not an expert in the field of dementia).

The researchers mention that they included people taking one of seven different statins (lines 87-88). However, in lines 343-344, they mention that further studies would be needed to examine the influence of specific statins on AD. Why was this influence not also studied for the statins used? These results would provide valuable information for follow-up studies.

In lines 106-108, the authors describe the matching criteria they used for conditional logistic regression. Since the primary purpose of matching is to prevent the control and AD groups from confounding effects with respect to important and known factors influencing the outcome variable, a rationale should be given for choosing exactly these four matching parameters.

Lines 144 + 145 have redundant content with lines 106 + 108, so you could delete the explanations in lines 106-108.
The confidence interval in line 211 is not identical to the CI shown in Figure 2.

Lines 290-298 are identical to lines 298-306, and this paragraph should also be corrected.

The description of the statistical analysis should be a bit more detailed. Because of the large number of covariates, a stepwise approach is recommended to identify the model with the best fit.
Possible additive and multiplicative interaction effects should also be discussed.

Round 2

Reviewer 1 Report

The authors resolved all my comments. I do not have any comments now.